# Screening and verification of reference genes for analysis of gene expression in winter rapeseed (*Brassica rapa* L.) under abiotic stress

**Li Ma**[1], **Junyan Wu**[1,2], **Weiliang Qi**[1,2], **Jeffrey A. Coulter**[3], **Yan Fang**[1], **Xuecai Li**[2], **Lijun Liu**[1], **Jiaojiao Jin**[1,2], **Zaoxia Niu**[1,2], **Jinli Yue**[1,2], **Wancang Sun**[1,2]*

**1** Gansu Provincial Key Laboratory of Aridland Crop Science, Gansu Agricultural University, Lanzhou, China, **2** College of Agronomy, Gansu Agricultural University, Lanzhou, China, **3** Department of Agronomy and Plant Genetics, University of Minnesota, St. Paul, MN, United States of America

* 18293121851@163.com

**Data Availability Statement:** All relevant data are within the manuscript and its Supporting Information files.

## Abstract

Winter rapeseed (*Brassica rapa* L.) is the main oilseed crop in northern China and can safely overwinter at 35 (i.e., Tianshui, China) to 48 degrees north latitude (i.e., Altai, Heilongjiang, Raohe, and Xinjiang, China). In order to identify stable reference genes to understand the molecular mechanisms of stress tolerance in winter rapeseed, internal reference genes of winter rapeseed under four abiotic stresses were analyzed using GeNorm, NormFinder, BestKeeper, and RefFinder software. The most stable combinations of internal reference genes were *β-actin* and *SAND* in cold-stressed leaves, *β-actin* and *EF1a* in cold-stressed roots, *F-box* and *SAND* in high temperature-stressed leaves, and *PP2A* and *RPL* in high temperature-stressed roots, *SAND* and *PP2A* in NaCl-stressed leaves, *RPL* and *UBC* in NaCl-stressed roots, *RPL* and *PP2A* in PEG-stressed leaves, and *PP2A* and *RPL* in PEG-stressed roots. Expression profiles of *PXG3* were used to verify these results. The stable reference genes identified in this study are useful tools for identifying stress-responsive genes to understand the molecular mechanisms of stress tolerance in winter rapeseed.

## Introduction

Abiotic stresses such as cold, heat, salt, and drought limit plant growth and yield. Plants have developed a variety of mechanisms to respond to the damage caused by these stresses, including complex series of transcriptions and regulation [1]. Real-time quantitative polymerase chain reaction (RT-qPCR) analysis is the most fundamental method for studying gene transcription and regulation, and it is extremely sensitive, specific, reproducible, and cost-effective [2]. However, there are often errors in application of RT-qPCR and interpretation of the results. One of the most common mistakes is inappropriate selection of reference genes for normalizing expression of the target gene [3, 4]. Ideally, reference genes are expressed at constant levels to represent the concentration of cDNA in a sample. However, their expression is

**Funding:** This study was financially supported by the Scientific research start-up funds for openly-recuited doctors of Gansu Agricultural University (GAU-KYQD-2019-17), the Utilization Technology of Rapeseed Heterosis and Creation of Strong Heterosis of China (2016YFD0101300), Agriculture Research System of China (CARS-12), and the Agriculture Research System of Gansu Province (GARS-TSZ-1), and the National Natural Science Foundation of China (31860388). The funders had no role in study design, data collection and analysis, decision to publish, or preparation of the manuscript.

**Competing interests:** The authors have declared that no competing interests exist.

usually altered by the effects of experimental treatment, tissue sites, and nucleic acid quality [5]. There are several commonly used reference genes in the literature, but the results often indicate that these used should be selected based on the species and experimental design [2, 6, 7]. In recent years, ICG (http://icg.big.ac.cn) has integrated more than 750 internal reference genetic studies (including 73 species of animals, 115 species of plants, 12 species of fungi, and 9 species of bacteria) to identify reference genes corresponding to specific experimental conditions [8]. Scientists have developed several methods for systematic verification of reference genes, such as NormFinder, Best-keeper, GeNorm, and RefFinder software, which integrate information on the expression of internal reference genes and measure their relative stability by sequencing [9–12]. The best method currently considered to follow the MIQE (Minimum Information for Publication of Quantitative Real-Time PCR Experiments) guidelines, using multiple internal reference genes, because the use of a single internal reference gene is considered inappropriate [13].

Internal reference gene screening has been conducted in different parts and tissues of multiple plant species subjected to a variety of stresses, including *Poa pratensis* L. [6], *Solanum tuberosum* L. [7], *Cynodon dactylon* L. [12], *Triticum Aestivum* L. [14], *Lactuca sativa* Linn. [15], and *Populus euphratica* [16]. In chicory (*Cichorium intybus* L.), Delporte demonstrated that *TIP41* (*TIP41*-like protein) was the most stable reference gene in cell cultures under various conditions, and these genes were validated in parallel on their seedlings, and result shown that the best reference gene is *Clath* (*Clathrin adapator complex subunit*) [17]. Wang et al. analyzed nine candidate reference genes in leaves of *Brassica napus* L. and found *TIP41* and *PP2A* to performed best and were applicable under various conditions [18]. However, *Brassica napus* and winter rapeseed have different ploidy, and their adaptability to environment and planting areas are obviously different, especially in China [19–21].

In China, winter rapeseed (*Brassica rapa L.*) is mainly produced in northern latitudes with harsh growing conditions and frequently experiences abiotic stresses such as cold, heat, drought, and salt [22–24]. This study screened and validated 10 internal reference genes of winter rapeseed under abiotic stresses to more accurately and extensively use RT-qPCR for gene analysis to understand the molecular mechanisms of abiotic stress tolerance in rapeseed.

## Methods

### Plant material and experimental treatment

In this study, winter rapeseed Longyou-7 (*Brassica rapa* L.) varieties widely cultivated in northern China were used. Plants were grown in a greenhouse at Gansu Agriculture University in Gansu Province (Lanzhou), China. During March to May 2018, 100 seeds were surface-sterilized in 10% $H_2O_2$ (hydrogen peroxide) for 30 min, soaked in distilled water for 10 min, and washed 3 times to remove $H_2O_2$. Seeds were germinated on two layers of wet filter paper in a glass petri dish and placed in a plant incubator (22°C with 16 h light/8 h dark cycle) for 2 days. Vigorous plants of the same growth stage were selected and transplanted into plastic seedling pots (mouth diameter × bottom inner diameter × height = 10 × 6 × 7 cm) containing 260 g of a 3:1 (volume:volume) ratio of matrix:vermiculite. One plant was grown in each pot and 200 ml of distilled water was added at the time of transplanting, with the same amount of distilled water added every 2 days. When the seedlings reached the one-leaf stage, 200 mL of 1/2 Hoagland nutrient solution and 200 mL of distilled water were added to each pot every 3 d. Plants were grown under normal conditions (22°C, 16 h light/8 h dark cycle) until the six-leaf stage, when plants of uniform size were selected and divided into four groups for application of four abiotic stresses. The drought stress treatments were imposed by adding 200 mL of a nutrient solution containing 18% PEG 6000 to pots [25], the salt stress treatments were imposed by

adding 200 mL of a nutrient solution containing 180 mM NaCl to pots [26]. The cold and heat stress treatments were applied in an incubator set at 4 and 40˚ C, respectively. Leaves and roots were harvested from all treatments following 0 (CK), 1, 3, 6, 12, 24 and 48 h of stress for all stress treatments, and immediately frozen in liquid nitrogen and stored at -80˚C for further analysis. Three separate pots were used as replicates for each stress treatment type and duration.

## Total RNA isolation, cDNA synthesis and qPCR

Total RNA was isolated from leaves and roots using the Plant Total RNA Extraction Kit (TaKaRa Biotechnology Co., Ltd., Dalian, China) according to the kit instructions. RNA concentration and mass were evaluated by detection of the A260/A280 and A260/A230 ratios, respectively, using a spectrophotometer (NanoVueTM plus, Wilmington, DE, USA). Genomic DNA contamination removal and first strand cDNA synthesis were performed using the PrimeScriptTM RT kit with gDNA Eraser (TaKaRa Biotechnology Co., Ltd., Dalian, China) according to the manufacturer's instructions. According to the standard that 10 μl reaction system can use up to 500 ng of total RNA, the RNA volume is calculated to ensure that all the added RNA is reverse transcribed into cDNA. Finally, we will dilute the cDNA concentration of all samples to 50 ng/μl according to the method described. The qPCR reaction conditions were as follows: 30 s at 95˚C, followed by 40 cycles of 5 s at 95˚C and 30 s at 60˚C, followed by 65–95˚C melting curve detection. The negative reverse transcriptase reaction (no reverse transcriptase) was carried out after DNA contamination was removed by gDNA eraser. The CT value obtained by fluorescence quantitative PCR was compared with that of the negative control.

## Candidate reference gene selection and primer design

Ten candidate reference genes (*ACT7*, *GADPH*, *TIP41*, *F-box*, *UBC*, *SAND*, *RPL*, *β-TUB*, *EF1α*, *PP2A*) have been reported in top 10 internal control genes ranked in ICG (http://icg.big. ac.cn/index.php/ICG:Statistics) and published papers for various plant species [18, 27, 28] and were considered to be potential candidate genes for this study. The coding sequence design of the candidate reference genes was designed using Primer Premier 5.0 (Premier, Canada) software according to the following parameters: melting temperature (Tm) of 58–62˚C, ideal Tm of 60˚C, GC content of 45–55%, ideal content of 50%; length of 17–24 bp, and amplicon length of 70–250 bp. Primer-specific assays using RT-PCR and pre-qPCR experiment with cDNA of CK, the product with a single expected size DNA band and a single peak in the RT-qPCR dissolution curve (Fig 1). The cDNA of CK was diluted five times in a 10-fold gradient and the slope of the standard curve was calculated after RT-qPCR using the LightCycle 96 (Roche, Basel, Switzerland) system. The amplification efficiency (E) of each reference gene was based on E (%) = [$10^{(-1/\text{slope})}$-1]×100 calculation [13, 29, 30]. Descriptions of candidate reference genes and primer sequences are shown in Table 1 and S1 Fig, and amplification efficiency was ≥96.3%.

## Stability analysis of internal reference gene expression

GeNorm [10], NormFinder [9], BestKeeper [11], and RefFinder software programs (http:// fulxie.0fees.us/?type=reference) were used to assess the stability of 10 potential reference genes. For the GeNorm and NormFinder programs, the original Cq (quantification cycle) value was converted to a relative Q value using the formula Q = $2^{-\Delta Cq}$, where ΔCq = each corresponding Cq value-minimum Cq value [31, 32]. The Q value was then uploaded in the GeNorm program and the expression stability measurement (M value) was calculated based on the

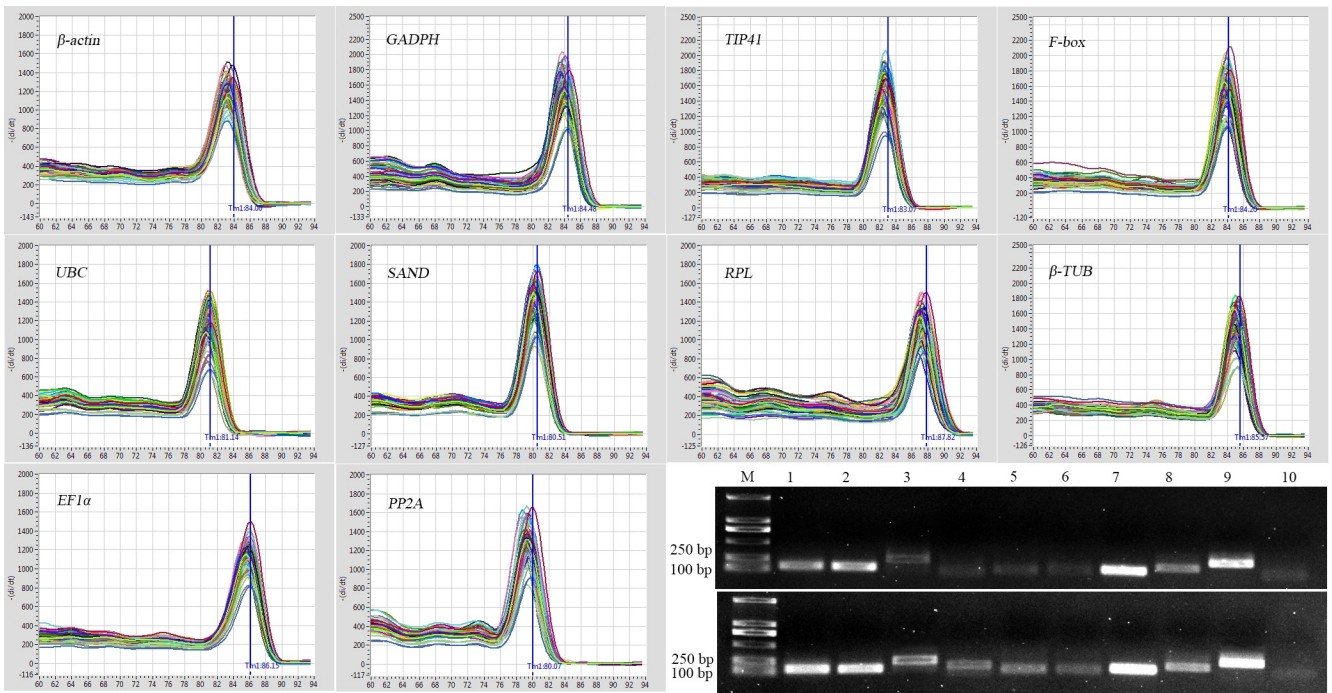

**Fig 1. Primer specificity and amplicon size.** Agarose gel (1.5%) electrophoresis indicated the amplification of a single PCR product for 10 genes (lines 1–10: *β-actin*, *GADPH*, *TIP41*, *F-box*, *UBC*, *SAND*, *RPL*, *β-TUB*, *EF1α*, and *PP2A*, respectively). The left gel is leaf and the right gel is root, M represents a 2000 bp DNA marker.

average of the pairwise variation of the candidate reference gene and all other detected genes. The NormFinder program uses an ANOVA-based model to calculate stability values to account for intra- and inter-group changes in candidate reference genes, with the lowest value

**Table 1. Description of primer sequences.**

| Gene symbol | Accession ID | Description | Primer sequence (5′-3′) (forward/reverse) | Amplicon length | Product Tm (˚C) | RT-qPCR efficiency (%) |
|---|---|---|---|---|---|---|
| *β-actin* | XM_018658258.2 | β-Actin-2 | TGTGCCAATCTACGAGGGTTT / TTTCCCGCTCGGCTGTTGT | 137 | 84 | 98.4 |
| *GADPH* | XM_009125769.3 | Glyceraldehyde-3-phosphate Dehydrogenase | CGTCCACTCCATCACTGC / AGAACCTTTCCGACAGCC | 132 | 84.4 | 103.7 |
| *TIP41* | XM_009116214.2 | TIP41-like protein | TAGCGGAGTTGTTGAGAAAG / AGCCAAAATCGTAAGAGGAG | 252 | 83.1 | 102.2 |
| *F-box* | XM_009153742.3 | F-box/kelch-repeat protein | GTCTGTCTTTATGCGGTCC / GATGCTCTCTCCCTCGTTC | 181 | 84.2 | 110.1 |
| *UBC* | XM_009136845.3 | Ubiquitin-conjugating enzyme | TACGCAGGAGGAGTGTTT / TGTTGATGTTTGGGTGGT | 106 | 81.1 | 96.3 |
| *SAND* | XM_009135631.3 | SAND family protein | ATACCGAGCATACCAGAA / GTGACCCAGCATAGCAGA | 108 | 80.5 | 101.6 |
| *RPL* | XM_009148505.3 | 60S Ribosomal protein L8 | CACTCACCACCGCAAGGGC / GGATGACGGAAGGAGACGC | 141 | 87.8 | 98.1 |
| *β-TUB* | XM_009125342.3 | Tubulin beta-4 | CTTGCTAATCCCACTTTTG / ACTGTTGCGACCCTCTTGA | 193 | 85.6 | 108.9 |
| *EF1α* | XM_009122323.2 | Elongation factor 1-alpha 1 | TGCTGTAACAAGATGGATG / CTGAAGTGGGAGACGGAGG | 267 | 86.2 | 99.9 |
| *PP2A* | XM_009120007.3 | Protein Phosphatase PP2A-2 | AGGGCTATCACCTTCTC / ACACATTGGTCCTTCGT | 85 | 80.1 | 105.2 |

representing the highest stability. For the BestKeeper program, the raw Cq values were used to calculate the coefficient of variation and standard deviation. The RefFinder program integrates results from the GeNorm (M value), NormFinder (stability value), BestKeeper (CV and SD) programs, along with ΔCq values, and generates a comprehensive ranking [6].

## Validation of reference gene stability

Previous studies have shown that *Probable peroxygenase 3* (*PXG3*) may respond to abiotic stress [29, 33]. In order to confirm the reliability of the selected reference genes, *PXG3* (Bra022936) was designed to design primers PXG3F: TTCAACCCAATCTCCTG, PXG3R: GTGATGGCAACCAACTC, and the relative expression profiles of Longyou-7 of leaves under drought and cold stress were detected and normalized. The most stable and unstable reference gene identified by the RefFinder program was used to calculate relative expression data using the $2^{-\triangle\triangle Cq}$ method [34] for three biological replicates.

## Results

### Expression level and variation of candidate reference genes

RT-qPCR detection of expression levels of 10 candidate reference genes showed that Cq values of all candidate reference genes under different treatments ranged from 16.8 to 29.1 (Fig 2). *EF1α* had the highest expression level, with an average Cq of 19.7, a range of 16.8–22.6, and a coefficient of variation of 7.6%. β-actin had an average Cq of 20.5 with the narrowest range

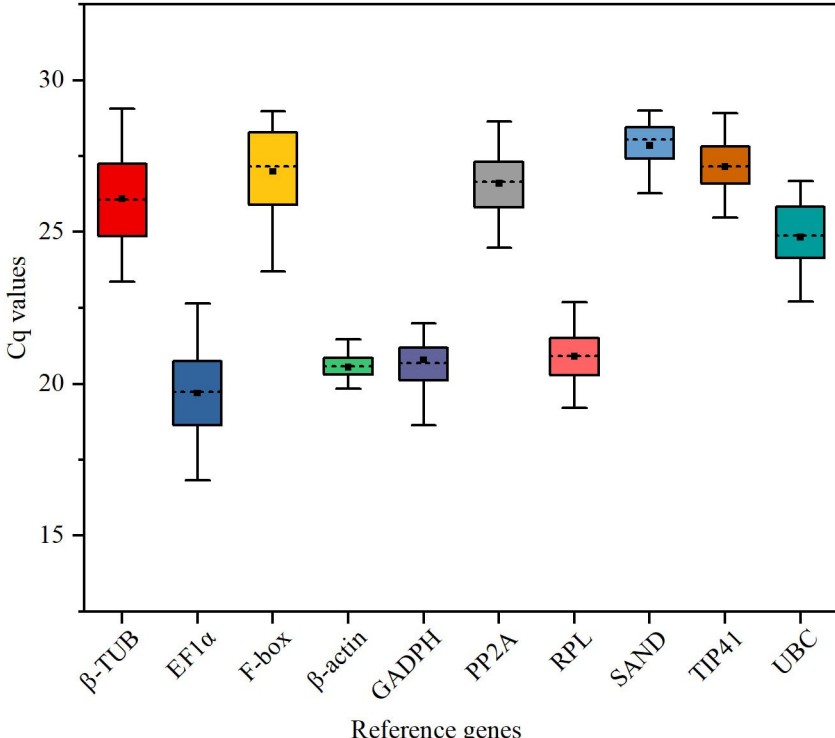

**Fig 2. Cq value of candidate reference genes across all treatments under four abiotic stresses.** The dashed horizontal line within a box-plot represents the median. The lower and upper edges of boxes show the 25th and 75th percentiles, respectively. Whiskers represent the maximum and minimum values.

among candidate reference genes (17.9–23.3), and a coefficient of variation of 5.4%. *SAND* and *TIP41* had the lowest expression levels, with an average Cq of 27.8 and 27.2, respectively, and a coefficient of variation of 2.6 and 3.3%, respectively. In all treatments, *EF1α* and *β-TUB* showed the greatest variability, with a coefficient of variation of 7.6 and 5.5%, respectively, while *PP2A*, *SAND* and *TIP41* exhibited the smallest variation, with a coefficient of variation of 3.7, 2.6, and 3.3%, respectively.

## Stability analysis of candidate reference genes using GeNorm software

The GeNorm software program was used to assess the stability of 10 candidate reference genes, which was defined as the average variation of one gene relative to all other genes. The threshold for eliminating stable genes was an average expression stability value (M) < 1.5, as a lower M value indicates higher stability [10]. Based on this criterion, *RPL* and *PP2A* were the most stable reference genes in pooled samples from all treatments (Total) and drought-treated roots (PR) (Fig 3), *β-actin* and *GADPH* were most stable in cold-treated leaves (CL), *β -actin* and *PP2A* were most stable in cold-treated roots (CR), *EF1a* and *UBC* were most stable in heat-treated leaves (HL), *PP2A* and *F-box* were most stable in heat-treated roots (HR), *PP2A* and *SAND* were the most stable in salt-treated leaves (NL), and *RPL* and *GADPH* were the most in drought-treated leaves (PL). *TIP41* was the most unstable reference gene in all samples (S1 File).

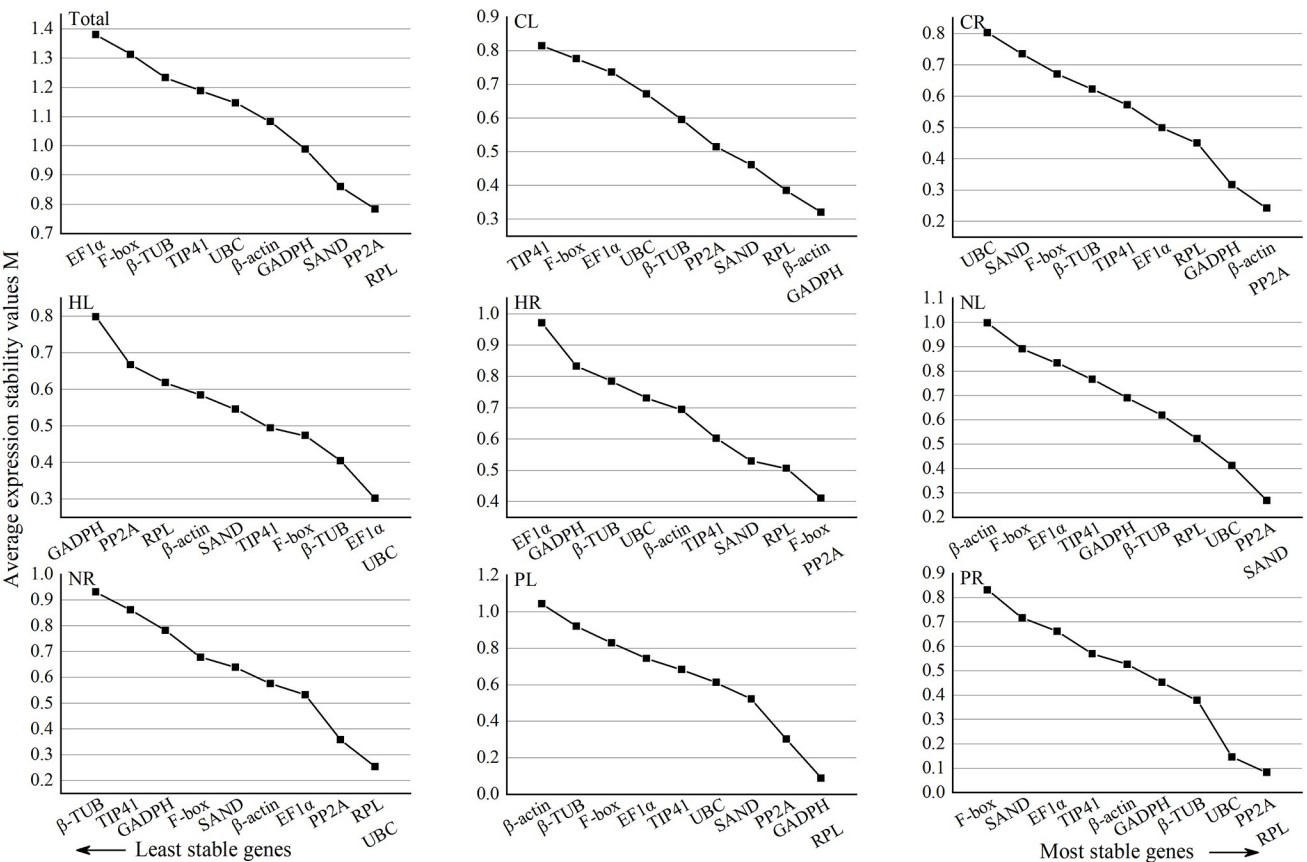

**Fig 3. Average expression stability (M) value of the 10 candidate reference genes assayed with GeNorm software.** Within a pane, the most and least stable genes are on the right and left, respectively. CL and CR: Cold-treated leaves and roots, respectively; HL and HR: Heat-treated leaves and roots, respectively; NL and NR: Salt-treated leaves and roots, respectively; PL and PR: Drought-treated leaves and roots, respectively; Total: Pooled samples from all treatments.

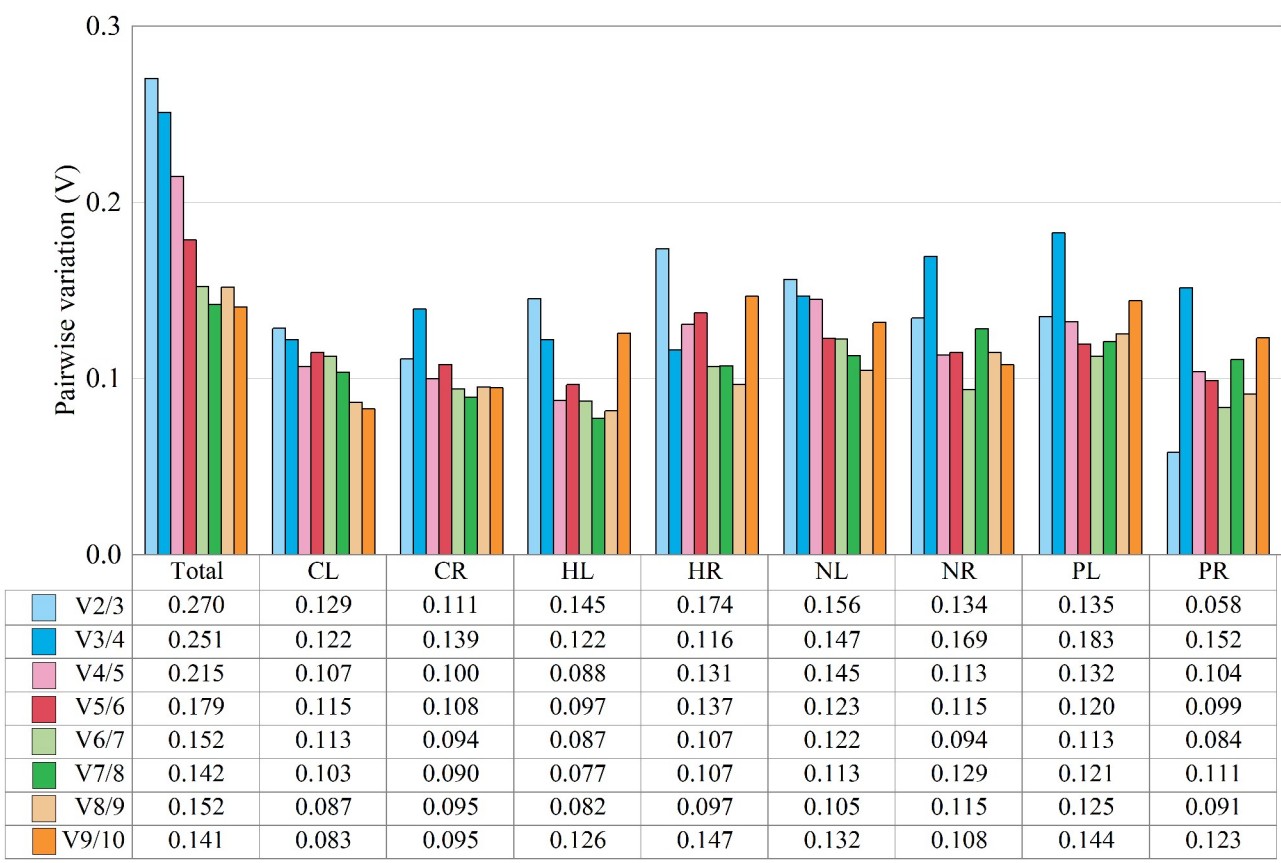

**Fig 4. Pairwise variation (V) of candidate reference genes.** The Vn/Vn+1 value is used to determine number of reference genes required for treatments; usually the value should be less than 0.15 for inclusion of a reference gene. CL and CR: Cold-treated leaves and roots, respectively; HL and HR: Heat-treated leaves and roots, respectively; NL and NR: Salt-treated leaves and roots, respectively; PL and PR: Drought-treated leaves and roots, respectively; Total: Pooled samples from all treatments.

Paired variant (Vn/Vn+1) values were also calculated using the GeNorm software program to determine the optimal number of reference genes required for RT-qPCR to normalize target gene expression levels [12]. A small change between Vn/Vn+1 and Vn+1/Vn+2 indicates that the addition of another reference gene has no significant effect on normalization, and a Vn/Vn+1 value of 0.15 is considered a threshold for determining whether to add a reference gene [10, 35]. The V2/3 values in CL (0.129), CR (0.111), HL (0.145), NR (0.134), PL (0.135), and PR (0.058) were lower than 0.15 (Fig 4), indicating that two reference genes were sufficient to normalize target gene expression (S1 File). The V2/3 values in HR (0.174) and NL (0.156) were higher than 0.15, and their V3/4 values were 0.116 and 0.147, respectively, indicating that three reference genes were required to normalize target gene expression. However, some researchers have suggested that '0.15' should not be considered as a strict threshold and that a higher Vn/Vn+1 threshold may be optimal in some cases [9, 36].

## Stability analysis of candidate reference genes using NormFinder software

Stability values of the 10 candidate reference genes were calculated using the NormFinder software program, and lower values represent higher stability. *RPL* was the most stable reference

gene in pooled samples from all treatments (Total, 0.343) and in NR (0.185), *SAND* was the most stable in CL (0.233) and NL (0.063), *EF1α* was the most stable in CR (0.085), *PP2A* was the most stable in PL and PR, F-box was the most stable in HR and the most unstable in PR, and *TIP41* was unstable for multiple treatments (Table 2 and S2 File). Consistent with the GeNorm anlaysis, *RPL* was the most stable gene in all samples; however, in most samples, the stability level of candidate reference genes generated with NormFinder was slightly different from that with GeNorm. For example, in the NormFinder analysis, *PP2A* and *SAND* were identified as the most stable reference genes in PL and CL, while the GeNorm analysis ranked these as third and fourth in stability, respectively.

## Stability analysis of candidate reference genes using BestKeeper software

The expression stability of 10 candidate reference genes was evaluated based on the Cq value calculated using the BestKeeper software program. The coefficient of variation and standard deviation of all candidate reference genes were calculated, and lower values of these indicate higher stability [37]. *β-actin* was the most stable reference gene in CL (2.1 ± 0.59), CR (0.52 ±0.11), and NR (0.48 ± 0.1), *PP2A* was the most stable in HL (1.97 ± 0.53) and PR (0.83±0.22), *SAND* was the most stable in pooled samples from all treatments (2.1 ± 0.59), *UBC* was the most stable in HR (0.68 ± 0.16), and *RPL* was the most stable in PL (0.38 ± 0.08) (Table 3 and S3 File). *β-TUB* was the most stable gene in NL, but was the most unstable in NR and PL. *EF1a* was the most unstable in pooled samples from all treatments, and in CL and HR, but ranked second in NL and NR, and *β-TUB* was unstable in most treatments. For most treatments, the stability ranking of candidate reference genes generated using BestKeeper was different from that with GeNorm and NormFinder.

## Ranking of candidate reference genes using RefFinder software

The RefFinder software program was used to determine the overall ranking of candidate reference genes. The program integrates the GeNorm, NormFinder, BestKeeper, and △Cq methods [12, 32, 38]. Under all treatments, the ranking order (from the most stable to the least stable) was: *RPL > SAND > PP2A > GADPH > β-actin > TIP41 > UBC > β-TUB > F-box > EF1α* (Table 4). Under CL treatments, the ranking order (from the most stable to the least stable) was: *β-actin > SAND > GADPH > RPL > β-TUB > PP2A > EF1α > UBC > F-box > TIP41*. Under CR treatments, the ranking order (from the most stable to the least stable) was:

**Table 2. Stability value of candidate reference genes determined using NormFinder software.**

| Rank | Total | CL | CR | HL | HR | NL | NR | PL | PR |
|------|-------|-----|-----|-----|-----|-----|-----|-----|-----|
| 1 | *RPL* (0.343) | *SAND* (0.223) | *EF1α* (0.085) | *SAND* (0.217) | *F-box* (0.13) | *SAND* (0.063) | *RPL* (0.185) | *PP2A* (0.14) | *PP2A* (0.028) |
| 2 | *SAND* (0.41) | *β-TUB* (0.294) | *GADPH* (0.241) | *F-box* (0.228) | *PP2A* (0.142) | *PP2A* (0.183) | *UBC* (0.237) | *RPL* (0.212) | *RPL* (0.028) |
| 3 | *GADPH* (0.537) | *β-actin* (0.322) | *TIP41* (0.323) | *RPL* (0.263) | *RPL* (0.31) | *UBC* (0.362) | *β-actin* (0.264) | *GADPH* (0.22) | *UBC* (0.088) |
| 4 | *PP2A* (0.613) | *RPL* (0.358) | *RPL* (0.334) | *TIP41* (0.307) | *SAND* (0.361) | *GADPH* (0.388) | *PP2A* (0.308) | *SAND* (0.354) | *TIP41* (0.358) |
| 5 | *TIP41* (0.667) | *GADPH* (0.372) | *β-actin* (0.371) | *EF1α* (0.33) | *TIP41* (0.389) | *RPL* (0.419) | *EF1α* (0.321) | *UBC* (0.414) | *EF1α* (0.437) |
| 6 | *β-actin* (0.708) | *PP2A* (0.426) | *F-box* (0.425) | *β-actin* (0.333) | *β-TUB* (0.506) | *TIP41* (0.479) | *SAND* (0.442) | *TIP41* (0.48) | *β-TUB* (0.455) |
| 7 | *UBC* (0.742) | *EF1α* (0.47) | *β-TUB* (0.436) | *UBC* (0.395) | *β-actin* (0.545) | *β-TUB* (0.56) | *F-box* (0.544) | *EF1α* (0.58) | *SAND* (0.479) |
| 8 | *β-TUB* (0.767) | *F-box* (0.482) | *PP2A* (0.439) | *β-TUB* (0.421) | *UBC* (0.581) | *EF1α* (0.597) | *GADPH* (0.66) | *β-TUB* (0.742) | *GADPH* (0.515) |
| 9 | *F-box* (0.911) | *UBC* (0.527) | *SAND* (0.544) | *PP2A* (0.472) | *GADPH* (0.648) | *F-box* (0.685) | *β-TUB* (0.732) | *F-box* (0.745) | *β-actin* (0.54) |
| 10 | *EF1α* (0.951) | *TIP41* (0.558) | *UBC* (0.645) | *GADPH* (0.865) | *EF1α* (1.007) | *β-actin* (0.901) | *TIP41* (0.736) | *β-actin* (0.987) | *F-box* (0.845) |

CL and CR: Cold-treated leaves and roots, respectively; HL and HR: Heat-treated leaves and roots, respectively; NL and NR: Salt-treated leaves and roots, respectively; PL and PR: Drought-treated leaves and roots, respectively; Total: Pooled samples from all treatments.

**Table 3. Stability of candidate reference genes determined using BestKeeper software.**

| Rank | Total | CL | CR | HL | HR | NL | NR | PL | PR |
|---|---|---|---|---|---|---|---|---|---|
| 1 CV ±SD | SAND 2.1±0.59 | β-actin 0.47 ±0.1 | β-actin 0.52 ±0.11 | PP2A 1.97 ±0.53 | UBC 0.68±0.16 | β-TUB 3.48 ±0.96 | β-actin 0.48 ±0.1 | RPL 0.38±0.08 | PP2A 0.83 ±0.22 |
| 2 CV ±SD | TIP41 2.72 ±0.74 | PP2A 1.13 ±0.29 | PP2A 0.94 ±0.24 | SAND 2.14±0.6 | β-actin 1.04 ±0.21 | EF1α 4.27±0.84 | EF1α 1.19±0.21 | GADPH 0.53 ±0.11 | UBC 1.05±0.25 |
| 3 CV ±SD | PP2A 3.01±0.8 | GADPH 1.31 ±0.27 | GADPH 1.44 ±0.3 | F-box 2.21 ±0.62 | TIP41 1.41 ±0.38 | F-box 3.5±0.96 | UBC 1.84±0.46 | PP2A 1.13±0.3 | GADPH 1.15 ±0.23 |
| 4 CV ±SD | RPL 3.26±0.68 | SAND 1.36 ±0.37 | TIP41 1.88 ±0.51 | TIP41 2.38 ±0.66 | PP2A 1.69 ±0.45 | β-actin 3.26 ±0.67 | PP2A 2.21±0.6 | SAND 2.21 ±0.62 | β-TUB 1.18 ±0.29 |
| 5 CV ±SD | β-actin 3.49 ±0.72 | RPL 1.54±0.32 | EF1α 1.95 ±0.36 | β-actin 2.67 ±0.6 | F-box 1.82 ±0.45 | GADPH 4.47 ±0.99 | GADPH 2.29 ±0.46 | UBC 2.66±0.65 | RPL 1.18±0.24 |
| 6 CV ±SD | UBC 3.68±0.92 | β-TUB 2.32 ±0.6 | F-box 2.18 ±0.56 | β-TUB 2.8 ±0.76 | GADPH 2.15 ±0.43 | PP2A 2.38 ±0.66 | RPL 2.29±0.48 | TIP41 2.83 ±0.77 | β-actin 1.42 ±0.29 |
| 7 CV ±SD | GADPH 3.74 ±0.78 | UBC 2.75±0.68 | β-TUB 2.27 ±0.57 | RPL 3.02±0.66 | SAND 2.66 ±0.74 | RPL 1.53±0.33 | TIP41 2.44 ±0.64 | β-actin 3.05 ±0.58 | TIP41 1.52 ±0.41 |
| 8 CV ±SD | F-box 4.5±1.21 | F-box 2.86±0.8 | SAND 2.77 ±0.76 | UBC 3.6±0.91 | RPL 3.47±0.74 | SAND 1.87 ±0.52 | F-box 3±0.76 | F-box 3.31 ±0.91 | SAND 2.56±0.7 |
| 9 CV ±SD | β-TUB 4.59 ±1.2 | TIP41 2.96 ±0.82 | RPL 2.87±0.58 | EF1α 4.41±0.89 | β-TUB 3.79 ±1.01 | TIP41 1.77 ±0.46 | SAND 3.01 ±0.82 | EF1α 3.51±0.71 | EF1α 3.26±0.63 |
| 10 CV ±SD | EF1α 6.08±1.2 | EF1α 3.72±0.72 | UBC 4.13 ±1.05 | GADPH 4.97 ±1.08 | EF1α 5.65±1.2 | UBC 2.88±0.73 | β-TUB 3.81 ±0.99 | β-TUB 3.7 ±0.93 | F-box 3.54 ±0.96 |

CV and SD: coefficient of variation and standard deviation, respectively; CL and CR: Cold-treated leaves and roots, respectively; HL and HR: Heat-treated leaves and roots, respectively; NL and NR: Salt-treated leaves and roots, respectively; PL and PR: Drought-treated leaves and roots, respectively; Total: Pooled samples from all treatments.

**Table 4. The most stable and unstable combinations of reference genes determined using RefFinder analysis.**

| Sample | Stability | Genes (ranking value) | | |
|---|---|---|---|---|
| Total | Most | RPL (1.19) | SAND (1.86) | PP2A (3.13) |
| | Least | EF1α (9.46) | F-box (9.24) | |
| CL | Most | β-actin (1.57) | SAND (2.11) | GADPH (2.66) |
| | Least | TIP41 (10) | | |
| CR | Most | β-actin (1.97) | EF1α (2.11) | GADPH (2.45) |
| | Least | UBC (10) | | |
| HL | Most | F-box (2.21) | SAND (2.45) | |
| | Least | GADPH (10) | | |
| HR | Most | PP2A (2.06) | RPL (2.82) | F-box (2.99) |
| | Least | EF1α (10) | | |
| NL | Most | SAND (1.32) | PP2A (2) | |
| | Least | F-box (9) | | |
| NR | Most | RPL (1.5) | UBC (1.86) | β-actin (2.78) |
| | Least | β-TUB (9.74) | | |
| PL | Most | RPL (1.41) | PP2A (1.73) | GADPH (2.06) |
| | Least | β-TUB (8.97) | | |
| PR | Most | PP2A (1.19) | RPL (1.57) | |
| | Least | F-box (10) | | |

CL and CR: Cold-treated leaves and roots, respectively; HL and HR: Heat-treated leaves and roots, respectively; NL and NR: Salt-treated leaves and roots, respectively; PL and PR: Drought-treated leaves and roots, respectively; Total: Pooled samples from all treatments.

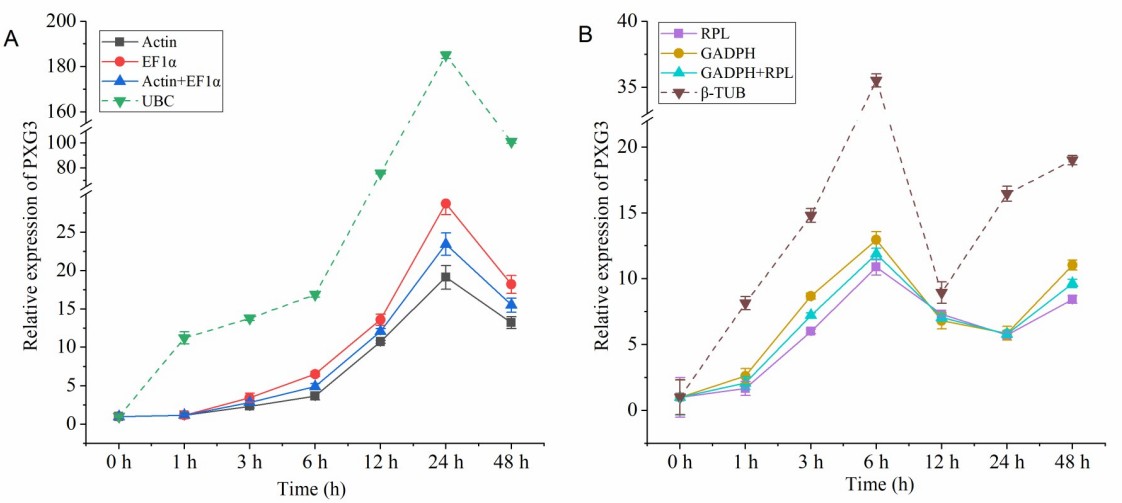

**Fig 5. Relative expression of the *PXG3* target gene for validation of selected reference genes.** (A) cold stress roots, (B) drought stress leaves.

*β-actin > EF1α > GADPH > PP2A > TIP41 > RPL > F-box > β-TUB > SAND > UBC. SAND* and *F-box*, were the most stable for HL, *PP2A* and *RPL* were the most stable for HR, *PP2A* and *SAND* were the most stable for NL, *RPL* and *UBC* were the most stable for NR, *PP2A*, *GADPH*, and *RPL* were the most stable for PL, and *PP2A* and *RPL* were the most stable for PR. *β-TUB*, *F-box*, and *EF1α* were the most unstable reference genes in most of the abiotic stress treatments (S1 Table).

## Expression of target genes and validation of selected reference genes

To verify the stability of the most stable and unstable reference genes obtained in the aforementioned analyses, the expression pattern of the target gene *PXG3* in CR and PL was analyzed (Fig 5 and S2 Table). In CR, the most stable reference genes were *β-actin* and *EF1α*, and the most unstable reference gene was *UBC* (Fig 5A). In PL, the most stable reference genes were *RPL* and *GADPH*, and the most unstable reference gene was *β-TUB* (Fig 5B). The expression of *PXG3* in the roots of winter rapeseed was slightly up-regulated after ≤6 h of cold stress, and significantly up-regulated after 12–24 h of cold stress (Fig 5A). When *UBC* was used as the reference gene, the expression was significantly up-regulated at 1 h of cold stress, the expression for all durations of cold stress was significantly higher than that of the other three reference genes. Therefore, the expression profile was more stable when the reference gene was combined. Under drought stress, *PXG3* in leaves of winter rapeseed was expressed at the highest level at 6 h of stress (Fig 5B). When using *GAPDH* and *RPL* alone, or *GAPDH* combined with *RPL* as the reference gene, the expression of *PXG3* was more stable and consistent than that with *β-TUB*, and the expression was most stable for the combined reference gene. When *β-TUB* was used as a reference gene, the expression of *PXG3* fluctuated greatly, especially after 6–48 h of drought stress.

## Discussion

RT-qPCR is widely recognized as a method for accurately and sensitively quantifying gene transcription levels, even for genes with lower transcription levels. For efficient RT-qPCR, precise standardization of gene expression is required for appropriate internal reference controls,

and most gene expression studies in the literature are typically standardized using a single internal reference control [39]. The validity of the experimental results depends to a large extent on the reference gene applied, so it is necessary to verify expression stability of the control gene under specific experimental conditions before using for standardization [14, 33]. The most commonly used reference genes in early gene expression studies were based primarily on their roles that are known or expected in basic cellular processes. The actin gene has been the most commonly used for quantification of normalized gene expression levels [40]. Genes encoding *GAPDH*, *actin*, and *EF1α* have been used as the most relevant reference genes for fruit development [41]. However, in subsequent studies it was found that the actin gene is not suitable as a broad reference gene because transcript levels are observed in different plant tissues and organs under different growth conditions [27].

Analyses with the GeNorm, NormFinder, and BestKeeper software programs are based on statistical principles and are used extensively in RT-qPCR experiments to identify the stability of reference genes. The NormFinder program works similarly to GeNorm program, but the GeNorm program can be used to screen for the best combination of reference genes and numbers. Compared to the GeNorm and NormFinder programs, the BestKeeper program directly uses the quantitative result of the Cq value calculation [11]. The results of this study show that the results obtained by the GeNorm and NormFinder programs were similar, but different from the results obtained with the BestKeeper program, concurrent with Rapacz et al. [42]. The RefFinder program was used for comprehensive comparative analysis of reference genes to determine the final stability ranking because it integrates the comprehensive evaluation of GeNorm, NormFinder, BestKeeper, and △Cq [43], and has been used in many other species [15, 30, 34, 39, 44, 45].

In this study we selected 10 *Brassica rapa* reference genes according to the internal reference gene TOP10 of plant species that are closely related to the internal control genes of *Brassica rapa* [3, 8, 16, 34, 46–49]. The reference genes identified differed with stress treatment and plant tissue. Of 13 reference genes previously selected for identification in *Brassica rapa* L. ssp. pekinensis subjected to various stresses, *UBC*, *EF1α*, and *β-actin* were recommended for abiotic stress induced by hormones, salt, drought, cold, and heat [45]; the results of this study are similar. Others found that if only one reference gene is used in Chinese cabbage research, *EF1α* is the best choice for standardization of different tissues, but for higher accuracy, the combination of *EF1α* and *Apr* should be considered to improve the normalization factor [50]. These researchers also concluded that *GAPDH* is the best single reference gene for experiments under conditions of drought stress and downy mildew infection, but the combination of *GAPDH* and *UBC* enhances the normalization factor. Our study found that *RPL* had a relatively high ranking in all treatments, while *GAPDH* had a relatively high ranking in some samples. In *Poa pratensis* L., a combination of *β-actin* and *RPL* were identified as stable reference genes in heat-treated roots; however, *RPL* was the most unstable in drought-treated leaves [6]. Other studies found *RPL17* to be the best reference gene for short-term salt and ABA stress in *Populus euphratica* [16, 51]. Our results add another possible stable reference gene (*RPL*) besides *β-actin* to *Brassica rapa*. Studies on *Brassica napus* found that the combinations of *PP2A* + *UBC* and *F-box* + *SAND* performed well under drought and cold stress, and that two new reference genes (*TIP41* and *PP2A*) were expressed under multiple abiotic stresses [18]. In this study, *PP2A* was stable in leaves and roots under drought stress, and *SAND* was stable in cold-stressed leaves. These are similar to the above results, but *TIP41* was not among the combinations of reference genes ranked as most stable for any treatment in this study, which may attributable to differences in species [14, 32, 36, 47, 52, 53]. *β-actin* and *GADPH* were stable internal reference genes in cold-stressed leaves and roots, and *RPL* and *UBC* were stable in drought-stressed roots.

The expression pattern of the target gene *PXG3* in response to cold and drought stress in winter rapeseed was analyzed and the reliability of the identified stable reference genes was verified. *EF1α* and *β-actin* or *EF1α* and *β-actin* were used as reference genes for cold-stressed root samples, and *PXG3* expression profiles in winter rapeseed were used when *GAPDH* and *RPL* or *GAPDH* combined with *RPL* were used as reference genes for drought-stressed leaf samples. These genes may be suitable for quantitative normalization of target gene expression profiles in cold-stressed roots or drought-stressed leaves of *Brassica rapa* L. When a reference gene with poor stability is selected, the *PXG3* expression spectrum fluctuates, indicating that these genes are not reliable for RT-qPCR analysis. Selecting unstable reference genes for RT-qPCR analysis can lead to inaccurate experimental conclusions [12, 14–16, 54]. Therefore, selection of the best reference gene is particularly important for standardization of RT-qPCR and the rational expression of target genes.

## Conclusions

In this study, the GeNorm, NormFinder, BestKeeper, and RefFinder programs were used to analyze the internal reference genes of four abiotic stresses in winter rapeseed. *β-actin* and *SAND* were the most stable internal reference gene combination in cold-stressed leaves, *β-actin* and *EF1α* were the most stable combination in cold-stressed roots. *F-box* and *SAND* were the most stable combination in heat-stressed leaves, *PP2A* and *RPL* were the most stable combination in heat-stressed roots. *SAND* and *PP2A* were the most stable combination in NaCl-stressed leaves, *RPL* and *UBC* were the most stable combination in NaCl-stressed roots. *RPL* and *PP2A* were the most stable combination in drought-stressed leaves, and *PP2A* and *RPL* were the most stable combination in drought-stressed roots. These were verified by *PXG3* expression profiles. It is also believed that *RPL* can be used as a universal reference gene for multiple abiotic stresses in *Brassica rapa*. The stable reference gene identified in this study are a useful tool for identifying stress-responsive genes and can advancing knowledge of the underlying molecular mechanisms of abiotic stress tolerance in rapeseed.

## Supporting information

**S1 Fig. RT-qPCR standard curve of the 10 reference genes.**
(TIF)

**S1 File. Stability analysis of candidate reference genes using GeNorm software.** CL and CR: Cold-treated leaves and roots, respectively; HL and HR: Heat-treated leaves and roots, respectively; NL and NR: Salt-treated leaves and roots, respectively; PL and PR: Drought-treated leaves and roots, respectively; Total: Pooled samples from all treatments.
(ZIP)

**S2 File. Stability analysis of candidate reference genes using NormFinder software.** CL and CR: Cold-treated leaves and roots, respectively; HL and HR: Heat-treated leaves and roots, respectively; NL and NR: Salt-treated leaves and roots, respectively; PL and PR: Drought-treated leaves and roots, respectively; Total: Pooled samples from all treatments.
(ZIP)

**S3 File. Stability of candidate reference genes determined using BestKeeper software.** CL and CR: Cold-treated leaves and roots, respectively; HL and HR: Heat-treated leaves and roots, respectively; NL and NR: Salt-treated leaves and roots, respectively; PL and PR: Drought-treated leaves and roots, respectively; Total: Pooled samples from all treatments.
(ZIP)

**S4 File. Sequencing results of 10 genes specifically amplified products.**
(ZIP)

**S1 Table. Ranking of candidate reference genes using RefFinder software.** CL and CR: Cold-treated leaves and roots, respectively; HL and HR: Heat-treated leaves and roots, respectively; NL and NR: Salt-treated leaves and roots, respectively; PL and PR: Drought-treated leaves and roots, respectively; Total: Pooled samples from all treatments.
(XLS)

**S2 Table. Relative expression of the *PXG3* target gene for validation of selected reference genes.** (A) cold stress roots, (B) drought stress leaves.
(XLSX)

**S1 Raw images. The original gel images contained in the manuscript's main figures.**
(PDF)

## Author Contributions

**Conceptualization:** Li Ma, Wancang Sun.

**Data curation:** Li Ma, Junyan Wu, Xuecai Li, Jiaojiao Jin, Zaoxia Niu, Jinli Yue.

**Methodology:** Weiliang Qi, Lijun Liu, Jinli Yue.

**Software:** Weiliang Qi, Yan Fang, Xuecai Li.

**Writing – original draft:** Li Ma, Junyan Wu, Jeffrey A. Coulter, Wancang Sun.

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
