## [Decision Letter · Decision Letter 0]

26 Mar 2020

PONE-D-20-00260

Screening and Verification of Reference Genes for Analysis of Gene Expression in Winter Rapeseed (Brassica rapa L.) under Abiotic Stress

PLOS ONE

Dear Prof. Sun,

Thank you for submitting your manuscript to PLOS ONE. After careful consideration, we feel that it has merit but does not fully meet PLOS ONE’s publication criteria as it currently stands. Therefore, we invite you to submit a revised version of the manuscript that addresses the points raised during the review process.

We would appreciate receiving your revised manuscript by May 10 2020 11:59PM. To enhance the reproducibility of your results, we recommend that if applicable you deposit your laboratory protocols in protocols.io, where a protocol can be assigned its own identifier (DOI) such that it can be cited independently in the future. For instructions see: http://journals.plos.org/plosone/s/submission-guidelines#loc-laboratory-protocols

We look forward to receiving your revised manuscript.

Kind regards,

Yong Pyo Lim

Academic Editor

PLOS ONE

Journal Requirements:

"This study was financially supported by the Utilization Technology of Rapeseed Heterosis and Creation of Strong Heterosis of China (2016YFD0101300), Agriculture Research System of China (CARS-12), and the Agriculture Research System of Gansu Province (GARS-TSZ-1), and the National Natural Science Foundation of China (31860388)."

"Specify the role(s) played."

Reviewers' comments:

Reviewer's Responses to Questions

**Comments to the Author**

1. Is the manuscript technically sound, and do the data support the conclusions?

Reviewer #1: Yes

Reviewer #2: Yes

2. Has the statistical analysis been performed appropriately and rigorously? 

Reviewer #1: No

Reviewer #2: Yes

3. Have the authors made all data underlying the findings in their manuscript fully available?

Reviewer #1: Yes

Reviewer #2: Yes

4. Is the manuscript presented in an intelligible fashion and written in standard English?

Reviewer #1: No

Reviewer #2: Yes

5. Review Comments to the Author

Reviewer #1: Comments for the author

This paper attempts to identify appropriate reference genes for gene expression studies in Brassica rapa. The stability of ten potential reference genes was evaluated in four different abiotic stresses, in leaves and root samples. The expression stability of these genes was analyzed using geNorm, NormFinder, BestKeeper, and RefFinder software in order to identify the best reference genes under given experimental conditions. However, the article could have been more significant if the authors conducted a similar analysis for many different developmental stages and other abiotic stresses. Since such studies are not new, and a lot of work has been carried out with a more in-depth analysis of a large number of samples and genes.

I have several comments that should be clarified, namely:

1- The authors should mention a justification for choosing these particular 10 candidate genes in the materials methods.

2- No justification is given for the use of neither the concentration of 180mM NaCl for salt treatment nor the percentage of 18% of PEG for the drought treatment. Have the authors done some previous studies that show that these concentrations are adequate to induce the stresses?

3- In the case of plants obtained from seeds in which the variability will certainly be high, three biological replicates may not be sufficient to guarantee a good sampling; five replicates should be the minimum.

4- After DNAse treatment I do not see strong evidence (sufficient evidence) that contaminating genomic DNA was removed. The methods described (agarose gel and spectrophotometer) do not address the removal of contamination. The authors need to at least conduct a negative Reverse Transcriptase reaction whereby DNAse digested RNA is subjected to a mock cDNA synthesis without Reverse Transcriptase. The resulting reaction is then used for PCR as usual. In this case, the Ct value for the negative reactions could be compared to control reactions as evidence for "successful digestion".

5- Total RNA extraction and cDNA synthesis (Line 82) It is not clear how much total RNA was used for cDNA conversion. The authors should provide this detail and also confirm whether the same amount of total RNA was used for cDNA conversion across all the samples.

6- Candidate reference gene selection and primer design (Line 96) It is mentioned that the gel pictures were analyzed through 1.5% agarose gel on the RT-PCR product with a single expected size DNA band to confirm the specificity of the PCR product. I am not sure if the authors actually mean that they confirmed the product by the predicted size. The best practice is to sequence the product to confirm the specificity of the reaction. The authors should clearly state that their assessment was based on the product size and single band.

8- It is not clear which cDNA was used to confirm primers specificity (lines 96-97). Please mention in the text.

9- It is not clear which cDNA samples were used to determine primers efficiencies (Table 1). Please mention in the text.

10- Inside Table 1 regarding primer pairs, the amplicons more than 180 to 200 bp is not suitable for Real-time analysis. It’s usually recommended to keep amplicon size small and if possible about the same length across all test genes. Is there any particular reason why some of these genes have big amplicons?

11- “Brassica rapa”, “reference gene” “gene expression” and “abiotic stress” keywords are redundant since already present in the title.

12- Additionally, the most stably expressed reference genes for each stress were used for accurate normalization of the expression level of Probable peroxygenase 3 (PXG3) in leaves of Longyou-7 under drought and cold stress.

It seems that one of the conclusions of the study was that none of the candidate reference genes were uniformly expressed across tissues and all the experimental conditions tested in this study.

13- Correct the nomenclature of Fig 5 in throughout the result. The author has written Fig 4 instead of Fig 5.

14- The authors should also, provide statistical analysis for data shown in fig 5.

15- The English writing needs to revise. Also, some correction should be done for some incorrect words inside the text, for example in conclusion (line 317) gene name is RPL, not PRL, so many extra spaces found across the manuscript for example line 75, 120, 162 etc. Inline 47 species name should be in italic Cynodon dactylon. Please rephrase line 276 to 296 in the discussion. In-text, many places it is NACL, it should be NaCl.

Reviewer #2: The authors of this ms have carried out similar approach to the previous one in B. napus (Wang et al. 2014). Normalization of expression levels of genes will be very important to predict their gene function. Therefore, development of appropriate reference genes can greatly contribute to the transcriptional regulation of gene expression. It is also true in the study of B. rapa crops. However, one should consider duplication and polyploidization of Brassicaceae. B. rapa represents one of the diploid genomes (with triplicated genome) that forms allopoloid B. napus. Therefore, Wang et al. (2014) reported that TIP41 is the best-ranked reference gene in B. napus under several stress conditions (two parolog gens are present in B. napus), but not in B. rapa (only one gene in the geneome). Paralog genes might differentially expressed in different tissues or different conditions, and expression levels will be sum of them, implying more paralogs better. Beta-actin-2 or -7 represents at least 4 paralogs in B, rapa. Therefore, authors should consider how many paralog genes are present in B. rapa genome for data explanation.

- Beta-actin -7 used by authors appears to be beta-actin-2. Please check gene and primer sequences in Table 1.

- References should be intensively edited: Capital vs. lowercase, removal of printing company, etc.

6. PLOS authors have the option to publish the peer review history of their article (what does this mean?). If published, this will include your full peer review and any attached files.

Reviewer #1: No

Reviewer #2: No

---

## [Author Response · Author response to Decision Letter 0]

8 May 2020

Response to Reviewer #1:

In China, winter rapeseed (Brassica rapa L.) is mainly produced in northern latitudes with harsh growing conditions and frequently experiences abiotic stresses such as cold, heat, drought, and salt. At present, about 115 kinds of plants have carried out the identification of internal reference genes. The results show that the internal reference genes identified by different stress treatments and tissue parts are different. However, this kind of research has not been carried out in winter rapeseed (Brassica rapa L.). This study is the first time to identify the reference genes of typical oil crops (winter Brassica rapa) in northern China, which provides support for the research on the mechanism and gene expression of cold resistance, drought resistance, salt resistance and heat resistance of winter rapeseed.

1- The authors should mention a justification for choosing these particular 10 candidate genes in the materials methods.

Response 1: Reasons for selecting these 10 candidate genes have been added to materials and methods.

2- No justification is given for the use of neither the concentration of 180mM NaCl for salt treatment nor the percentage of 18% of PEG for the drought treatment. Have the authors done some previous studies that show that these concentrations are adequate to induce the stresses?

Response 2: Yes, our research group has done these researches before. 180mm NaCl and 18% PEG are the critical concentrations of salt and drought stress in winter Brassica rapa. These results have been published and two references have been added to the manuscript.

Dong X, Mi C, Liu Z, et al. Response of winter rapessed seeding growth and physiological characteristics under PEG drought tolerance. J Henan Agric Univ. 2018;52: 313–321. 

Wang Z, Liu Z, Sun W, et al. Effects of NaCl and Na2SO4 stress on germination of winter rapeseed (Brassica rapa L.) and analysis of salt resistance. Agric Res Arid Areas. 2016;34: 243–252. 

3- In the case of plants obtained from seeds in which the variability will certainly be high, three biological replicates may not be sufficient to guarantee a good sampling; five replicates should be the minimum.

Response 3: For the experiment operation, including seed selection, pot experiment, growth management, abiotic stress and experiment operation, we carried out three biological repeats in strict accordance with the same standard, and the previous transcriptome, microRNA and fluorescence quantitative experiments got good results. Therefore, three biological repeats are feasible in winter Brassica rapa.

4- After DNAse treatment I do not see strong evidence (sufficient evidence) that contaminating genomic DNA was removed. The methods described (agarose gel and spectrophotometer) do not address the removal of contamination. The authors need to at least conduct a negative Reverse Transcriptase reaction whereby DNAse digested RNA is subjected to a mock cDNA synthesis without Reverse Transcriptase. The resulting reaction is then used for PCR as usual. In this case, the Ct value for the negative reactions could be compared to control reactions as evidence for "successful digestion".

Response 4: Yes, we used gDNA eraser to remove DNA pollution, and then we carried out negative reverse transcriptase reaction (without reverse transcriptase), and then we carried out fluorescence quantitative PCR to get CT value similar to that of negative control.

5- Total RNA extraction and cDNA synthesis (Line 82) It is not clear how much total RNA was used for cDNA conversion. The authors should provide this detail and also confirm whether the same amount of total RNA was used for cDNA conversion across all the samples.

Response 5: In the process of RNA extraction and cDNA reverse transcription, we strictly followed the instructions of the kit. After the completion of RNA extraction, RNA concentration, a260/A280 and a260/A230 were tested, and we will re extract the samples with non-standard results. According to the instructions of cDNA reverse transcription Kit (10 μl reaction system can use up to 500 ng of total RNA), the RNA volume is calculated to ensure that all the added RNA is reverse transcribed into cDNA. Finally, we will dilute the cDNA concentration of all samples to 50 ng/μl according to the method described. This detail has been added to the manuscript.

6- Candidate reference gene selection and primer design (Line 96) It is mentioned that the gel pictures were analyzed through 1.5% agarose gel on the RT-PCR product with a single expected size DNA band to confirm the specificity of the PCR product. I am not sure if the authors actually mean that they confirmed the product by the predicted size. The best practice is to sequence the product to confirm the specificity of the reaction. The authors should clearly state that their assessment was based on the product size and single band.

Response 6: Sorry, I didn't write the pre experiment steps in the manuscript. We will design 3-4 pairs of primers according to the principle of fluorescence quantitative primer design, and we will test the amplification effect and specificity of primers in multiple software before synthesis. After obtaining the primers, RT-PCR amplification and detection in 1.5% agarose, the most important thing is that we will carry out pre experiment on the primers, and determine the final primers after obtaining the ideal CT value, amplification curve and dissolution curve. All the experimental steps and questions in the manuscript will be communicated with the technical personnel of the kit manufacturer to get the best experimental results.

8- It is not clear which cDNA was used to confirm primers specificity (lines 96-97). Please mention in the text.

Response 8: The cDNA of CK was selected for primer specific detection. It has been mentioned in the text.

9- It is not clear which cDNA samples were used to determine primers efficiencies (Table 1). Please mention in the text.

Response 9: The cDNA of CK was selected for detection of primer efficiency. It has been mentioned in the text.

10- Inside Table 1 regarding primer pairs, the amplicons more than 180 to 200 bp is not suitable for Real-time analysis. It’s usually recommended to keep amplicon size small and if possible about the same length across all test genes. Is there any particular reason why some of these genes have big amplicons?

Response 10: After we designed the primers, we did not find the ideal pairing primers of TIP41 and EF1α genes within 200 bp by software and qPCR pre experiment analysis. Then we slightly increased the amplification length of the primers, and got the ideal pairing primers through the above methods, and there are similar reports in the related internal reference gene identification articles. 

11- “Brassica rapa”, “reference gene” “gene expression” and “abiotic stress” keywords are redundant since already present in the title.

Response 11: Key words have been modified in the manuscript.

12- Additionally, the most stably expressed reference genes for each stress were used for accurate normalization of the expression level of Probable peroxygenase 3 (PXG3) in leaves of Longyou-7 under drought and cold stress. It seems that one of the conclusions of the study was that none of the candidate reference genes were uniformly expressed across tissues and all the experimental conditions tested in this study.

Response 12: Ideally, reference genes are expressed at constant levels to represent the concentration of cDNA in a sample. However, their expression is usually altered by the effects of experimental treatment, tissue sites, and nucleic acid quality [5]. The best method currently considered to follow the MIQE (Minimum Information for Publication of Quantitative Real-Time PCR Experiments) guidelines, using multiple internal reference genes, because the use of a single internal reference gene is considered inappropriate [13]. 

In China, winter rapeseed (Brassica rapa L.) is mainly produced in northern latitudes with harsh growing conditions and frequently experiences abiotic stresses such as cold, heat, drought, and salt. At present, about 115 kinds of plants have carried out the identification of internal reference genes. The results show that the internal reference genes identified by different stress treatments and tissue parts are different, but this kind of research has not been carried out in in winter rapeseed (Brassica rapa L.).

13- Correct the nomenclature of Fig 5 in throughout the result. The author has written Fig 4 instead of Fig 5.

Response 13: It has been revised in the manuscript.

14- The authors should also, provide statistical analysis for data shown in fig 5.

Response 14: Statistical analysis has been added to figure 5.

15- The English writing needs to revise. Also, some correction should be done for some incorrect words inside the text, for example in conclusion (line 317) gene name is RPL, not PRL, so many extra spaces found across the manuscript for example line 75, 120, 162 etc. Inline 47 species name should be in italic Cynodon dactylon. Please rephrase line 276 to 296 in the discussion. In-text, many places it is NACL, it should be NaCl.

Response 15: Sorry, these errors have been corrected in the manuscript, and the full text has been carefully revised and checked.

Response to Reviewer #2:

The authors of this ms have carried out similar approach to the previous one in B. napus (Wang et al. 2014). Normalization of expression levels of genes will be very important to predict their gene function. Therefore, development of appropriate reference genes can greatly contribute to the transcriptional regulation of gene expression. It is also true in the study of B. rapa crops. However, one should consider duplication and polyploidization of Brassicaceae. B. rapa represents one of the diploid genomes (with triplicated genome) that forms allopoloid B. napus. Therefore, Wang et al. (2014) reported that TIP41 is the best-ranked reference gene in B. napus under several stress conditions (two parolog gens are present in B. napus), but not in B. rapa (only one gene in the geneome). Paralog genes might differentially expressed in different tissues or different conditions, and expression levels will be sum of them, implying more paralogs better. Beta-actin-2 or -7 represents at least 4 paralogs in B, rapa. Therefore, authors should consider how many paralog genes are present in B. rapa genome for data explanation.

- Beta-actin -7 used by authors appears to be beta-actin-2. Please check gene and primer sequences in Table 1.

- References should be intensively edited: Capital vs. lowercase, removal of printing company, etc.

Response Reviewer #2: There are different ploidy between Brassica rapa and Brassica napus, which shows that the internal reference genes need to be selected according to different species and stress treatment, there are more suitable reference genes than TIP41 in Brassica rapa. We used NCBI and Ensemble Genomes to compare the sequence of the internal reference gene, fully considered the existence of Paralog genes in Brassica rapa, and got the appropriate results.

We rechecked the gene and primer sequence in Table 1, and we used beta-actin-7.

We carefully revise the references and the full text to ensure correct writing.

---

## [Decision Letter · Decision Letter 1]

27 May 2020

PONE-D-20-00260R1

Screening and Verification of Reference Genes for Analysis of Gene Expression in Winter Rapeseed (Brassica rapa L.) under Abiotic Stress

PLOS ONE

Dear Dr. Sun,

Thank you for submitting your manuscript to PLOS ONE. After careful consideration, we feel that it has merit but does not fully meet PLOS ONE’s publication criteria as it currently stands. Therefore, we invite you to submit a revised version of the manuscript that addresses the points raised during the review process.

We look forward to receiving your revised manuscript.

Kind regards,

Yong Pyo Lim

Academic Editor

PLOS ONE

Reviewers' comments:

Reviewer's Responses to Questions

**Comments to the Author**

1. If the authors have adequately addressed your comments raised in a previous round of review and you feel that this manuscript is now acceptable for publication, you may indicate that here to bypass the “Comments to the Author” section, enter your conflict of interest statement in the “Confidential to Editor” section, and submit your "Accept" recommendation.

Reviewer #1: All comments have been addressed

Reviewer #2: All comments have been addressed

2. Is the manuscript technically sound, and do the data support the conclusions?

Reviewer #1: Yes

Reviewer #2: Partly

3. Has the statistical analysis been performed appropriately and rigorously? 

Reviewer #1: Yes

Reviewer #2: (No Response)

4. Have the authors made all data underlying the findings in their manuscript fully available?

Reviewer #1: Yes

Reviewer #2: Yes

5. Is the manuscript presented in an intelligible fashion and written in standard English?

Reviewer #1: Yes

Reviewer #2: No

6. Review Comments to the Author

Reviewer #1:

I found the manuscript has been improved following revision. However, there is still scope to improve it which affects manuscripts quality.

- Authors did not mentioned qPCR conditions in material and methods.

- A sequence of PCR product is also needed in order to confirm that the specific gene has been amplified.

- Please avoid some unnecessary some space in sentences and typing errors

For example line 52, 54, 56, 76 and 77 have spacing problem.

Line 26 there you typed EF-1a instead of EF1α.

Line 114 you added 2.4 as a bullet numbering.

Reviewer #2: I am sorry to say that author did not consider my first comments. I have checked most genes whether they have multigenes showing high identities. Except TIP4l, most genes have more than two paralogs with high sequence identities. Therefore, authors should check whether primer sequences shown in Table 1 is specific for one of them or common. XM_009127096.2 is beta-actin 7 gene, but it does not have primer sequences in Table 1. Instead, beta-actin 2 gene contains forward primer, but no reverse primer sequence at all. Please check, specify primer and paralog gene.

7. PLOS authors have the option to publish the peer review history of their article (what does this mean?). If published, this will include your full peer review and any attached files.

Reviewer #1: Yes: Sonam Singh

Reviewer #2: No

---

## [Author Response · Author response to Decision Letter 1]

26 Jun 2020

Response to Reviewer #1:

1- Authors did not mentioned qPCR conditions in material and methods.

Response 1: The reaction conditions of qPCR have been supplemented in material and methods.

2- A sequence of PCR product is also needed in order to confirm that the specific gene has been amplified.

Response 2: We rechecked 10 gene sequences with primers, A single band was obtained by PCR amplification with primers, and the specific sequence was obtained by sequencing. Here is the result.

β-actin: Sorry, Professor, we sequenced the actin primers to get a single strip and sequencing result. The comparison result is shown in the figure below. After NCBI comparison, it shows that it is beta actin 2 gene (XM_018658258.2), and the primer information has been updated.

GADPH: No paralogs with high sequence identities, and NCBI has provided a new version of this gene, which I have updated in materials and methods.

TIP41: A single band was obtained by PCR amplification with primers, and the specific sequence was obtained by sequencing. And NCBI has provided a new version of this gene, which I have updated in materials and methods.

F-box: A single band was obtained by PCR amplification with primers, and the specific sequence was obtained by sequencing. And NCBI has provided a new version of this gene, which I have updated in materials and methods.

UBC: A single band was obtained by PCR amplification with primers, and the specific sequence was obtained by sequencing. And NCBI has provided a new version of this gene, which I have updated in materials and methods.

SAND: A single band was obtained by PCR amplification with primers, and the specific sequence was obtained by sequencing. And NCBI has provided a new version of this gene, which I have updated in materials and methods.

RPL: A single band was obtained by PCR amplification with primers, and the specific sequence was obtained by sequencing. And NCBI has provided a new version of this gene, which I have updated in materials and methods.

β-TUB: A single band was obtained by PCR amplification with primers, and the specific sequence was obtained by sequencing. And NCBI has provided a new version of this gene, which I have updated in materials and methods.

EF1α: A single band was obtained by PCR amplification with primers, and the specific sequence was obtained by sequencing. And NCBI has provided a new version of this gene, which I have updated in materials and methods.

PP2A: A single band was obtained by PCR amplification with primers, and the specific sequence was obtained by sequencing. And NCBI has provided a new version of this gene, which I have updated in materials and methods.

3- Please avoid some unnecessary some space in sentences and typing errors

For example line 52, 54, 56, 76 and 77 have spacing problem.

Line 26 there you typed EF-1a instead of EF1α.

Line 114 you added 2.4 as a bullet numbering.

Response 3: These errors have been corrected in the manuscript and the full text examined carefully.

Response to Reviewer #2:

I am sorry to say that author did not consider my first comments. I have checked most genes whether they have multigenes showing high identities. Except TIP4l, most genes have more than two paralogs with high sequence identities. Therefore, authors should check whether primer sequences shown in Table 1 is specific for one of them or common. XM_009127096.2 is beta-actin 7 gene, but it does not have primer sequences in Table 1. Instead, beta-actin 2 gene contains forward primer, but no reverse primer sequence at all. Please check, specify primer and paralog gene.

Response Reviewer #2: Sorry, Professor, I didn't understand your question and didn't answer it completely in the first reply. This time, We rechecked 10 gene sequences with primers, and sequenced them after PCR amplification to ensure that the primers were specifically amplified. Here is the result.

β-actin: Sorry, Professor, we sequenced the actin primers to get a single strip and sequencing result. The comparison result is shown in the figure below. After NCBI comparison, it shows that it is beta actin 2 gene (XM_018658258.2), and the primer information has been updated.

GADPH: No paralogs with high sequence identities, and NCBI has provided a new version of this gene, which I have updated in materials and methods.

TIP41: A single band was obtained by PCR amplification with primers, and the specific sequence was obtained by sequencing. And NCBI has provided a new version of this gene, which I have updated in materials and methods.

F-box: A single band was obtained by PCR amplification with primers, and the specific sequence was obtained by sequencing. And NCBI has provided a new version of this gene, which I have updated in materials and methods.

UBC: A single band was obtained by PCR amplification with primers, and the specific sequence was obtained by sequencing. And NCBI has provided a new version of this gene, which I have updated in materials and methods.

SAND: A single band was obtained by PCR amplification with primers, and the specific sequence was obtained by sequencing. And NCBI has provided a new version of this gene, which I have updated in materials and methods.

RPL: A single band was obtained by PCR amplification with primers, and the specific sequence was obtained by sequencing. And NCBI has provided a new version of this gene, which I have updated in materials and methods.

β-TUB: A single band was obtained by PCR amplification with primers, and the specific sequence was obtained by sequencing. And NCBI has provided a new version of this gene, which I have updated in materials and methods.

EF1α: A single band was obtained by PCR amplification with primers, and the specific sequence was obtained by sequencing. And NCBI has provided a new version of this gene, which I have updated in materials and methods.

PP2A: A single band was obtained by PCR amplification with primers, and the specific sequence was obtained by sequencing. And NCBI has provided a new version of this gene, which I have updated in materials and methods.

---

## [Decision Letter · Decision Letter 2]

10 Jul 2020

Screening and Verification of Reference Genes for Analysis of Gene Expression in Winter Rapeseed (Brassica rapa L.) under Abiotic Stress

PONE-D-20-00260R2

Dear Dr. Sun,

We’re pleased to inform you that your manuscript has been judged scientifically suitable for publication and will be formally accepted for publication once it meets all outstanding technical requirements.

Kind regards,

Yong Pyo Lim

Academic Editor

PLOS ONE

Additional Editor Comments (optional):

Reviewers' comments:

Reviewer's Responses to Questions

**Comments to the Author**

1. If the authors have adequately addressed your comments raised in a previous round of review and you feel that this manuscript is now acceptable for publication, you may indicate that here to bypass the “Comments to the Author” section, enter your conflict of interest statement in the “Confidential to Editor” section, and submit your "Accept" recommendation.

Reviewer #1: All comments have been addressed

Reviewer #2: All comments have been addressed

2. Is the manuscript technically sound, and do the data support the conclusions?

Reviewer #1: Yes

Reviewer #2: Yes

3. Has the statistical analysis been performed appropriately and rigorously? 

Reviewer #1: Yes

Reviewer #2: Yes

4. Have the authors made all data underlying the findings in their manuscript fully available?

Reviewer #1: Yes

Reviewer #2: (No Response)

5. Is the manuscript presented in an intelligible fashion and written in standard English?

Reviewer #1: Yes

Reviewer #2: Yes

6. Review Comments to the Author

Reviewer #1: The comments were satisfactory answered in revised version of manuscript. It can be accepted for publication

Reviewer #2: Authors do their best to improve ms. Now this ms can be published in PLoS One, thereby helping someones who have studying transcripts profiles

7. PLOS authors have the option to publish the peer review history of their article (what does this mean?). If published, this will include your full peer review and any attached files.

Reviewer #1: No

Reviewer #2: No

---

## [Editor Report · Acceptance letter]

14 Jul 2020

PONE-D-20-00260R2 

Screening and Verification of Reference Genes for Analysis of Gene Expression in Winter Rapeseed (Brassica rapa L.) under Abiotic Stress 

Dear Dr. Sun:

I'm pleased to inform you that your manuscript has been deemed suitable for publication in PLOS ONE. Congratulations! Your manuscript is now with our production department. 

Kind regards, 

on behalf of

Dr. Yong Pyo Lim 

Academic Editor

PLOS ONE